# A Rare Pathological Phenotype of Endometrioid Serous and Clear-Cell Ovarian Cancer with PIK3CA Mutations in Relation to The Excellent Response of Alpelisib

**DOI:** 10.3390/genes14081632

**Published:** 2023-08-16

**Authors:** Ertugrul Bayram, Ghanim Khatib, Burak Guney, Emine Kilicbagir, Huru Rabia Gulec, Ibrahim Boga, Semra Paydas

**Affiliations:** 1Department of Medical Oncology, Cukurova University Faculty of Medicine, Adana 01250, Turkey; sepay@cu.edu.tr; 2Department of Obstetrics and Gynecology, Cukurova University Faculty of Medicine, Adana 01250, Turkey; ghanim.khatib@gmail.com; 3Department of Nuclear Medicine, Cukurova University Faculty of Medicine, Adana 01250, Turkey; isaburak@gmail.com; 4Department of Pathology, Cukurova University Faculty of Medicine, Adana 01250, Turkey; eminebagir@yahoo.com; 5Department of Biotechnology, Cukurova University Institute of Natural and Applied Sciences, Adana 01250, Turkey; rbglc6@gmail.com; 6AGENTEM (Adana Genetic Diseases Diagnosis and Treatment Center) and Medical Genetics Department of Medical Faculty, Cukurova University, Adana 01250, Turkey; ibr.boga@gmail.com

**Keywords:** alpelisib, ovarian cancer, PI3K inhibitor

## Abstract

Patients with metastatic ovarian cancer who develop resistance to standard therapy with or without platinum need to search for other therapeutic choices. Therefore, identifying genetic alterations and selecting an approach to treatment using precision medicine techniques are important. In a patient diagnosed with mixed-type ovarian cancer after surgery, adjuvant therapy was applied with a combination of carboplatin and taxane, but the disease recurred. Upon evaluation of the patient as having platinum-sensitive epithelial ovarian cancer (EOC), combination therapy with bevacizumab was initially successful. However, disease progression was again observed, and molecular analysis revealed the presence of an E545K mutation in the PIK3CA gene; therefore, a selective PI3K inhibitor, alpelisib, was used as a treatment under the compassionate-use protocol. The patient’s complications improved after receiving the alpelisib medication. The patient has been in complete remission for over two years. This case serves as a rare example that confirms the utility of alpelisib in managing mixed-type ovarian cancer.

## 1. Introduction

Among the gynecological cancers, ovarian cancer is the second-most prevalent and most lethal type. In Turkey, it is the seventh-most common cancer among women [1,2,3]. It is heterogeneous disease with varying prognoses among its various subgroups. Epithelial carcinomas (EOC) are the most prevalent kind of ovarian cancer, and high-grade serous ovarian cancer (HGSOC) is the most common and lethal subtype of EOC. Just nearly half of patients with this disease survive for more than five years [3,4]. Due to the lack of a specific screening test and the fact that the symptoms are mixed with other non-malignant conditions, the disease is diagnosed in the advanced stages and challenging to cure. CA125, a tumor marker for OC, is a routinely used blood test for screening. The disease’s progression is affected by a number of factors [1,2,3,4].

Presently, surgery and following adjuvant chemotherapy (CT) are the standard of care (SOC) in epithelial ovarian cancer (EOC). Choosing clinical trials of novel therapies and using predictive biomarkers that will allow patient selection for chemotherapy, immunotherapy or targeted drugs are necessary for achieving personalized precision medicine. Immunotherapy with checkpoint inhibitors with/without platinum-based CT as well as molecularly targeted treatments is important for precision medicine cases which can be used in relapsed/refractory epithelial ovarian cancer (RREOC). For this aim, exploring molecular changes causing relapse and disease metastasis is important in daily practice. Comprehensive molecular profiling is incorporated into present studies in order to better understand the difficulties associated with treating ovarian cancer. Accessing suitable evidence-based care will help to increase survival [4,5].

A PI3K-AKT-mTOR signaling pathway is important for cell survival and proliferation. It has been revealed that this pathway is related to cancer development and treatment resistance, as well as mechanisms including metastasis and angiogenesis in many types of cancer [6,7]. The PI3K subunit p110a protein is encoded by a mutated PIK3CA gene. Many cancers including ovarian, breast, head and neck squamous cell carcinoma, bladder, and colon cancer have been linked to PIK3CA mutations. Three hotspot mutations in PIK3CA were detected in human malignancies: E542K, E545K, and H1047R. Inhibition of the PI3K pathway is the treatment of choice in cases with PIK3CA mutations. The intracellular functions of insulin are mediated by PI3Ks. The metabolism of glucose is disrupted when the PIK3CA-encoded enzyme, p110, is selectively inhibited. When liver-specific PI3K mutant animals had reduced insulin sensitivity and increased gluconeogenesis, it was understood that p110 played an important role in these processes [7,8,9,10].

Alpelisib is a small, orally accessible PI3K inhibitor that selectively inhibits p110 about 50 times stronger than other isoforms do and is used in cases with PIK3CA mutations [10,11]. Exhibited early signs of optimistic activity in a population of advanced solid tumors that contained PIK3CA mutations and alpelisib–fulvestrant therapy significantly improved progression-free survival and overall response, based on a phase 3, randomized, double-blind trial (SOLAR-1). Alpelisib was approved by the FDA as a therapy for patients with hormone receptor-positive (HR+) and human epidermal growth factor receptor 2-negative (HER2-) PIK3CA-mutant breast cancer. When used in combination with other chemotherapy drugs as a co-adjuvant, it showed increased beneficial effects, in particular for breast and lung, HNSCC, and pancreatic cancer [6,10].

Since the main component of the regulation of glucose metabolism is the intracellular signaling mechanism known as the PI3K pathway and alpelisib prevents insulin’s systemic activity inside cells by inhibiting PI3K, the most common side effects were hyperglycemia, alopecia and maculopapular rash. Hyperglycemia is an adverse effect that reduces the effectiveness of the therapy and raises the incidence of treatment interruptions, dose reductions, and delays [10,11]. Herein, we present a case of R/R EOC treated with alpelisib with ongoing remission for about two years.

## 2. Case Presentation

A 44-year-old woman presented with a mass in the ovary. Total abdominal hysterectomy with bilateral salpingo-ophorectomy and bilateral pelvic paraaortic lymph node dissection was performed in June 2015. Mixed-type over carcinoma combined with high-grade serous ovarian cancer (HGSOC), endometrioid, and clear-cell carcinoma was reported and four cycles of adjuvant CT were performed with carboplatin (175 mg/m^2^) and taxane (350 mg/m^2^) every 3 weeks (August 2015).

The patient’s overall health status was frequently evaluated using the complete blood count, liver function tests, kidney function tests, and electrolyte levels. CA 125 levels were monitored over time to assess the patient’s response to therapy and the efficacy. Radiological tests were performed regularly in order to monitor tumor growth and detect possible metastatic lesions.

She presented with recurrent disease in September 2018, and a whole-body positron emission tomography/computed tomography (PET-CT) scan was revealed. The serum CA 125 value was found to be 81.3 U/mL. Metastatic lymph nodes were observed in the iliac region and biopsy mixed-type over carcinoma (HGSOC, endometrioid, and clear-cell carcinoma). The patient was evaluated as having platinum-sensitive EOC, a platinum, taxane, and bevacizumab (10 mg/kg) combination was administered for 6 cycles, and then the serum CA 125 value was normal. Maintenance bevacizumab was given in November 2019. The serum CA 125 increased to 56.4 U/mL while remission wasbeing monitored. A whole-body positron emission tomography/computed tomography (PET-CT) scan revealed the left iliopsoas muscle and external iliac region. The biopsy result from the metastatic lesion was reported as mixed-type ovarian cancer (HGSOC and endometrioid and clear-cell carcinoma) (Figure 1). At this point, she was evaluated as having platinum-refractory epithelial ovarian cancer (PREOC).

Next-generation sequencing (NGS) of a multigene panel including BRCAs and the PIK3CA gene was performed with a paraffin block, and a E545K mutation in the PIK3CA (NM_006218.3: c.1624G > A) gene was detected. Alpelisib treatment was started (300 mg daily) in November 2020 with the compassionate use program. Serum CA 125 levels returned to normal (22.9) (Figure 2), radiological regression was detected (Figure 3) in the lymph nodes, and the patient’s complaints were resolved after alpelisib treatment. Side effects and changes in glucose levels were monitored. Within the first 6 weeks of treatment, the patient exhibited a grade 1 cutaneous rash and mucositis. Antihistamines were given to the patient as a premedication before each chemotherapy session in order to treat the side effects that had been noticed. There was no significant alteration in blood glucose levels necessitating the use of medication. She has been followed for more than two years with complete remission under the alpelisib treatment.

This study was conducted in accordance with the Declaration of Helsinki. The patient has given written informed consent for the publication of the details about this case.

## 3. Discussion

OC is the seventh-most common cancer among women in Turkey [3] and is typically diagnosed in an advanced stage as having endometrioid, serous, and CCC (clear-cell carcinoma) histologies. Despite ongoing research and advancements, it remains the most fatal cancer in women. Understanding the pathophysiology of recurring ovarian cancers and identifying potential molecular targets for treatment are urgently needed due to the absence of efficient approaches for prevention and treatment. Cancer is a condition where the genome undergoes various dynamic alterations throughout time [2,9].

The management of RREOC is challenging from surgical and medical perspectives. Despite the majority of patients responding to platinum-based CT, disease which resistant/refractory to platinum is an important problem in clinical practice, and the timing of earlier therapy is an important factor that affects the choice of the following treatment. The medicines used to treat ovarian cancer that is resistant to cisplatin and carboplatin are doxorubicin, gemcitabine and bevacizumab. The duration of therapies following a relapse increases the toxicity, drug resistance, and cost. The studies conducted to lower the morbidity rate aimed to achieve the best outcomes with a smaller dose of drugs [12,13].

New approaches for treatment include biological techniques, hormone therapy, and stem cell therapy. Immunotherapy, antibody therapies, vaccines, inhibitors, and gene therapy are examples of biological treatments that can be grouped together. Each type of treatment has benefits and drawbacks of its own. Treatment approaches also differ from person to person. This situation proves that cancer is an individual disease. As a result, it is impossible to discuss a single, conclusive treatment for cancer. Either alone or in combination, the right methods of therapy can be used. At this point, NGS becomes a must in daily routines for molecular oncology practice to detect targets, and molecular targeted treatments are being developed to try to fulfill this critical therapeutic need [7,14].

The majority of oncogenes reside within a complex genomic landscape that is defined by other changes that could themselves be targeted or change how responsive a patient is to therapy. Future research for OC management should also focus on genes that inhibit the formation of tumors. These genes include those that function in the signal pathway. Moreover, in the next coming years, genome-wide sequencing analyses will come into practice to assess the genetic changes affecting cancer behavior and treatment options. Prescriptive and future studies on OC should include targeted therapies (apoptosis, signaling pathways, etc.), modulating DNA repair, silencing gene expression, and combining therapies of drugs and small molecules that can enhance the effectiveness of treatment while minimizing drug resistance [6,7,14]. PI3K pathway activation has a role in the development and growth of tumors as well as resistance to cancer treatments. PIK3CA mutations are important in the etiology of ovarian CCC and are linked to poor prognosis [9,15,16]. According to many studies, endometrial cancer has more phosphatidylinositol 3-kinase (PI3K) pathway signaling alterations than any other cancer does. Following PTEN, PIK3CA is the key gene that has been significantly altered. For the treatment of EC, PI3K/mTOR inhibitors, either alone or in combination, are the subject of numerous registered clinical trials [17].

The PIK3CA gene encodes heterodimeric lipid kinases, which control metastasis, apoptosis, and cell division. PI3Ks are crucial lipid kinases acting as major regulators of cell metabolism and survival [18,19] and treatment, hence focusing on the PI3K signaling pathway is essential in these cases. AKT and mTOR, two serine/threonine protein kinases, act as the pathway’s mediators. The PI3K/AKT/mTOR pathway is formed by these three parts, so it is ideal to create small-molecule inhibitors and several drugs are currently being tested in clinics. It is also important to consider their potential toxicity to normal cells [20,21].

Only p110 is the target of PI3K inhibitors. Dual PI3K and mTOR inhibitors are also accessible since the catalytic domains of p110 and mTOR share structural similarities [22]. Chemotherapy resistance could be brought on via the dysregulation of the PI3K/AKT/mTOR pathway. It is crucial to discover new therapeutics that focus on this route for the aforementioned reasons [23,24].

Numerous phase 1 research was conducted to investigate various alpelisib therapeutic dose regimens for cancer patients. Existing clinical trials and case reports primarily discussed alpelisib’s pharmacological effects on breast cancer [25,26]. A more accurate conclusion will be made using data from other current studies. As can be seen in Table 1, studies have shown that alpelisib, when used in combination with endocrine therapy, is successful in treating breast and ovarian cancer. Herein, we described an unusually long-term response of about 2 years to alpelisib in a case of RREOC with a CC histology showing a PIK3CA mutation. Alpelisib is a novel theraphy, and as drugs for the treatment of metastatic ovarian cancer continue to develop, practitioners must be conscious of any new possible adverse reactions that patients may experience; sometimes, the use of the medicine must cease immediately. In our case, alpelisib was well-tolerated, with grade 1 toxicity observed. The most common side effects of alpelisib are rash, hyperglycemia, and mucositis, which occurred within the first 6 weeks and were all manageable when the patient used antihistamines as a premedication. A similar exceptional response was reported in the other case, which provides conclusive clinical evidence for the effectiveness of alpelisib in the treatment of EC cancer patients with a PIK3CA mutation [17]. In another study using alpelisib, the objective response rate (ORR) of recurrent/advanced cervical cancer patients carrying the PIK3CA mutation was 33% [27].

Several cancers show PIK3CA gene mutations and these mutations make them therapeutic targets [26,30,31]. To better understand the potential mechanisms, more research is still needed. Our present case, with the first real-life data on alpelisib treatment in mixed-type over carcinoma, is unique because of the long-term response to alpelisib.

## 4. Conclusions

In conclusion, alpelisib is an example of how targeted therapies can be tailored to specific patient populations based on their genomic profiles in terms of precision medicine. Larger studies are needed to find patients with specific mutations and improve clinical outcomes.

## Figures and Tables

**Figure 1 genes-14-01632-f001:**
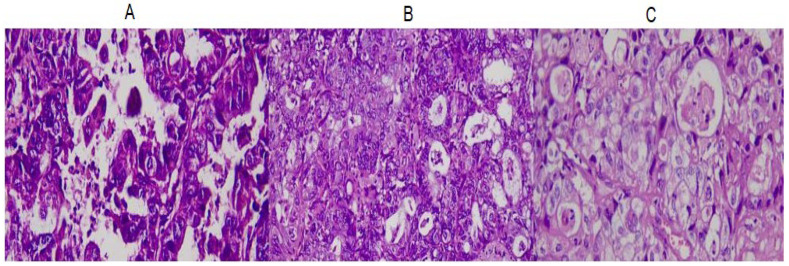
Images from the original biopsy of the patient, with ovarian mixed carcinoma. The images contain areas of serous carcinoma ((**A**), H&Ex400), endometrioid carcinoma ((**B**), H&Ex400), and clear-cell carcinoma ((**C**), H&Ex400).

**Figure 2 genes-14-01632-f002:**
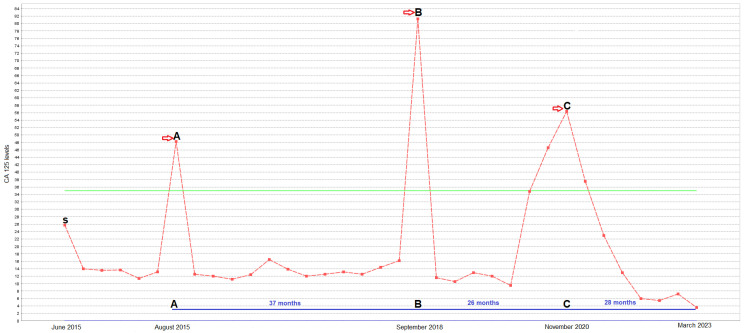
Changes in patient CA 125 levels (U/mL) during treatment (dates denoted by a red arrows). S: date of surgery and diagnosis. A: initiation of post-op adjuvant treatment (platinum and taxane). B: first-line chemotherapy after relapse (platinum, taxane, and bevacizumab). C: alpelisib therapy as second-line therapy after progression.

**Figure 3 genes-14-01632-f003:**
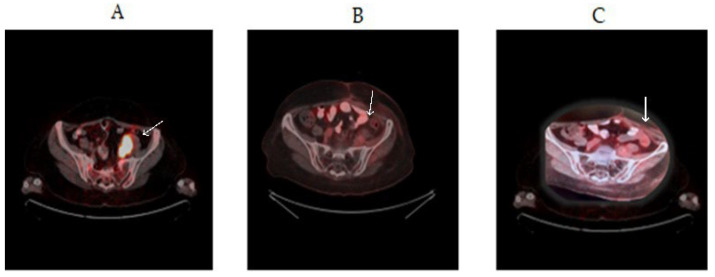
White arrows show lesion sites (**A**) Metastatic lesion measuring 3 cm adjacent to the sacrum and left ileopsoas muscle (in September 2018). (**B**) Complete response at 6 months after alpelisib treatment. (**C**) Complete response and remission in the 2nd year after alpelisib treatment.

**Table 1 genes-14-01632-t001:** Trials using alpelisib for ovarian cancer treatment and our case.

Research (NCT Number)	Disease	Targetable Mutations	Treatment	Response
Multicenter, open-label, phase Ib trial [28], NCT01623349	recurrent ovarian, fallopian tube, primary peritoneal, or breast cancer	*BRCA*	The starting oral dose level: 1 × 250 mg/day alpelisib + 2 × 100 mg/day olaparib. Level 1: 1 × 250 mg/day alpelisib + 2 × 200 mg/day olaparib. Level 2: 1 × 300 mg/day alpelisib + 2 × 200 mg/day olaparib. Level 3: 1 × 200 mg/day alpelisib + 2 × 200mg/day olaparib.	The response rate of olaparib and alpelisib was 30% in all patients with germline *BRCA* mutations and 33% in those who also had platinum-resistant disease.
Randomized, open-label, multicenter phase III trial [29], NCT04729387	platinum-resistant or -refractory high-grade serous ovarian cancer	no germline *BRCA* mutation	Oral 1 × 200 mg/day alpelisib and 2 × 200 mg/day olaparib.	In patients with platinum-resistant or –refractory, the combination of alpelisib/olaparib may offer improved efficacy compared to that of single-agent cytotoxic chemotherapy.
Phase II, NCT05238831	Locally advanced ovarian cancer		Oral 2 × 200 mg/day alpelisib.	Study results not yet published.
Our present case-study	Mixed-type ovarian cancer	*PIK3CA* (*E542K*)	Oral 2 × 200 mg/day alpelisib.	CR (complete response) over 2 years.

## Data Availability

The data that support the findings of this case report are available from the corresponding author upon reasonable request.

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
