# Peer review of "A Rare Pathological Phenotype of Endometrioid Serous and Clear-Cell Ovarian Cancer with PIK3CA Mutations in Relation to The Excellent Response of Alpelisib"

_genes, 2023, doi:10.3390/genes14081632_

Round 1
Reviewer 1 Report
This article is interesting and well written. It is well documented case of a rare mixed type of ovarian cancer with PIK3CA mutation and an excellent response of Alpelisib.
I suggest to put microphotos (Fig. 1) under high microscopic magnification (400x) - it is not possible to see features of tumour cells on present microphotographs.
Author Response
Dear Reviewer,
We would like to thank you for the suggestions and comments. We made changes that were detailed below. All revisions and improvements are highlighted yellow in the revised version of our manuscript. We would like to thank you again for your valuable time and comments for strengthen our paper.
Best regards,
New photos added but 400x is not avaliable.

Reviewer 2 Report
Overall, this is an interesting case study of PIK3CA mutation clear-cell ovarian cancer that was successfully treated with Alpelisib. It presents interesting context regarding the usefulness and success of whole genome sequencing and personalised therapy based on knowledge of the disease pathology. The article provides compelling evidence for the use of such techniques in ovarian cancer treatment. However, the publication suffers from the major pitfalls of all case studies, in that this is a singular case, and how practical the approach might be in generalised situations is unaddressed. Further information and data collection on the topic is definitely required for more robust conclusions and future clinical implementation, however, this article does provide proof of principle.
Some minor language issues which could be fixed, but overall does not affect the presentation of the contents.
Author Response
Dear Reviewer,
We would like to thank you for the suggestions and comments. We made changes that were detailed below. All revisions and improvements are highlighted yellow in the revised version of our manuscript. We would like to thank you again for your valuable time and comments for strengthen our paper.
Best regards,
Future prospective studies are required to confirm the effectiveness and safety of PIK3-inhibitor therapy in patients with an activating PIK3CA mutation.

Reviewer 3 Report
Bayram et al present an interesting case report on a patient with mixed-type ovarian cancer that initially responds to platinum chemotherapy but relapses after around 3 years. The patient responds again to platinum chemotherapy combined with bevacizumab. After 2 years, the patient relapsed again. Next generation sequencing found a mutation in PIK3CA and Alpelisib has been used successfully to put the patient in remission for 2 years. This is a nice study to show the use of NGS to identify targeted therapy for relapsed/refractory patients.
There are a few minor fixes that would help the paper. The figures and legends could be expanded/modified for clarity. In the legend for figure 1, it may be helpful to state that these images are from the original biopsy of the patient. It is in the text, but also helpful to be in the figure legend.
Figure 2 should have a Y-axis label. Also, it would be useful to mention what the first data point corresponds to
Figure 3, it would be helpful to have an arrow to highlight the area to focus on for non-radiologists who might read the paper. Also, it would also be beneficial to mention when the first PET data is from in the legend just to clarify.
There are a number of typos that need to be fixed, particularly mixed-type over carcinoma, which is written throughout.
Author Response
Dear Reviewer,
We would like to thank you for the suggestions and comments. We made changes that were detailed below. All revisions and improvements are highlighted yellow in the revised version of our manuscript. We would like to thank you again for your valuable time and comments for strengthen our paper.
Best regards,
We stated in the figure 1 legend that these images are from the original biopsy of the patient. Y-axis label added to figure 2. Additionally, we described what the first data point indicates. We added arrows to the figure 3 to show the area. In the legend, we stated when the first data came from.
